# Preparation of Highly Crystalline Silk Nanofibrils and Their Use in the Improvement of the Mechanical Properties of Silk Films

**DOI:** 10.3390/ijms231911344

**Published:** 2022-09-26

**Authors:** Ji Hye Lee, Bo Kyung Park, In Chul Um

**Affiliations:** 1Department of Biofibers and Biomaterials Science, Kyungpook National University, Daegu 41566, Korea; 2Buildings and Transportation Science Division, Oak Ridge National Laboratory, One Bethel Valley Road, Oak Ridge, TN 37831, USA

**Keywords:** silk nanofibril, composite film, mechanical properties, degumming, ultrasonication

## Abstract

Due to their commendable biocompatibility, regenerated silk fibroin (RSF) films have attracted considerable research interest. However, the poor mechanical properties of RSF films have limited their use in various biomedical applications. In this study, a novel, highly crystalline silk fibril was successfully extracted from silk by combining degumming with ultrasonication. Ultrasonication accelerated the development of silk nanofibrils measuring 130–200 nm on the surface of the over-degummed silk fibers, which was confirmed via scanning electron microscopy. Additionally, the crystallinity index of silk fibril was found to be significantly higher (~68%) than that of conventionally degummed silk (~54%), as confirmed by the Fourier-transform infrared (FTIR) spectroscopy results. Furthermore, the breaking strength and elongation of the RSF film were increased 1.6 fold and 3.4 fold, respectively, following the addition of 15% silk nanofibrils. Thus, the mechanical properties of the RSF film were remarkably improved by the addition of the silk nanofibrils, implying that it can be used as an excellent reinforcing material for RSF films.

## 1. Introduction

Silk is a natural composite consisting of fibroin and sericin. Because the silk has good blood compatibility [1,2], excellent cyto-compatibility [3,4], a low inflammatory response in the body [5], and good mechanical properties [6], extensive research has been conducted on its biomedical applications, such as artificial eardrums [7], membranes for guided bone regeneration [8,9], artificial heart valves [10], and nerve conduits [11].

Although an artificial eardrum manufactured with regenerated silk fibroin (RSF) film has been commercialized, the poor mechanical properties of RSF film have limited its use in other biomedical applications. The RSF film is too brittle and has a low elongation (<5%) because the highly crystallized silk structure is disrupted and the molecular weight of SF is decreased during degumming and dissolution of silk [12,13,14,15]. Therefore, various components have been added to RSF in numerous studies to improve its mechanical properties [16,17,18,19]. Although this is a simple strategy, there is a possibility that the foreign polymers deteriorate the unique properties of silk, such as its biocompatibility.

Nanofibrils have been used as a reinforcement filler in numerous materials. When it interacts with the surrounding polymers (matrix), it exhibits unique interactions with the matrix, resulting in outstanding toughness and strength. For instance, cellulose nanofiber is a highly crystallized form of cellulose, and its use as a reinforcement filler to improve the mechanical properties of the polymeric film has increased significantly over time [20,21,22].

In recent years, extensive studies have been conducted on the preparation of silk nanofibrils. Zhao et al. produced silk nanofibrils via ultrasonication of silk fibers in water [23]. Meanwhile, Zhang et al. fabricated a silk nanofibril film by the direct dissolution of silk fibers in a CaCl_2_-formic acid solution [24]. Additionally, Ling et al. fabricated silk nanofibril membranes using hexafluoroisopropanol (HFIP) in conjunction with ultrasonication treatments [25,26]. Moreover, Lv et al. fabricated silk nanofibrils using urea along with ultrasonication treatments, yielding a hybrid of silk fibril and metal–organic frameworks [27].

Silk nanofibrils can be used as reinforcement fillers to enhance the mechanical properties of RSF films. Because the nanofibrils are derived from silk fibers, the unique properties of silk are retained. However, silk nanofibrils have not yet been used to enhance the mechanical properties of RSF films by regulating their concentration. Although Zhang et al. directly prepared RSF films with silk nanofibrils by dissolving silk fibers in a CaCl_2_-formic acid solution [24], the silk nanofibrils were neither extracted nor added to the RSF film. Thus, controlling the amount of silk nanofibrils in RSF films is extremely challenging.

The current study reports the fabrication of silk nanofibrils using a novel approach. We over-degummed the silk to generate nanofibrils on the SF fibers and ultrasonically treated the degummed silk (i.e., SF) fibers to extract the silk nanofibrils, as opposed to using various chemical solvents (e.g., CaCl_2_, HFIP, and urea). Subsequently, different amounts of silk nanofibrils were added to the RSF solution to fabricate a silk nanofibril/RSF composite film with enhanced mechanical properties. In addition, we investigated the structural characteristics and properties of the silk nanofibril and the impact of silk nanofibril quantity on the silk nanofibril/RSF composite film.

## 2. Results and Discussion

### 2.1. Structural Characteristics of the Degummed and Ultrasonicated Silk

Degumming, a well-known method for removing the sericin layer from silk, has been accomplished in the past using various chemicals and techniques. In the current study, we controlled the concentration of sodium carbonate not only to completely remove sericin but also to separate the silk nanofibrils from the SF filament. Silk nanofibrils were observed on the surface of over-degummed SF filaments in a previous study [28]. When silk is degummed, sodium carbonate acts chemically, while the thermal energy and the movement of water molecules from boiling water physically attack the silk fibrils on the SF filament. Consequently, silk fibrils form on SF filaments. However, as shown in Figure 1A, the appropriate amount of sodium carbonate is essential to obtain the maximum number of fibrils from SF. As the amount of degumming agent increases, so does the degumming ratio. Considering that the silk cocoons used in this study have a sericin content of 26–27% [28,29] and the degumming ratios are greater than 27%, it can be concluded that the silk was over-degummed. Figure 1A shows that in the over-degummed silk, the sericin is completely removed and the SF is also partially damaged (i.e., silk fibrils are peeled off from SF filament).

When the sodium carbonate concentration is too low (<0.75%), sodium carbonate is capable of removing the silk sericin layer. However, the concentration is not sufficient to affect the SF filament. As the concentration of sodium carbonate exceeds 0.75%, the outer surface of the SF filament is exposed to the sodium carbonate and silk fibrils begin to separate. Accordingly, some silk fibrils are peeled off from the SF filaments, resulting in the presence of silk fibrils on the surface of the SF filament; some silk fibrils are peeled off excessively, resulting in their separation from the SF filament. Additionally, when the sodium carbonate content is too high (>1%), all split fibrils are removed from the surface of the SF filament, leaving no fibrils on the surface. Thus, the constant increase of the degumming ratio with increasing sodium carbonate content (Figure 1A) confirms the assumption that the fibrils are removed from the SF filaments by over-degumming. Moreover, as the amount of degumming agent was increased, the surface of the SF filament became rougher, indicating that the fibrils separated from the SF filaments. However, when sodium carbonate is added excessively (>1.25%), silk fibrils are completely peeled off the SF filaments and no silk fibrils are visible on the SF filament, as shown in Figure 1A(g). This indicates that a sodium carbonate concentration of 0.75% is optimal for obtaining the maximum silk fibrils from the SF filament with minimum loss.

To gain a better understanding of the generation and removal of nanofibrils in SF, silk fibers that were degummed using varying concentrations of sodium carbonate were analyzed using FTIR spectroscopy. As illustrated in Figure 1B, the degummed silk fibers demonstrated IR absorption at 1620, 1510, and 1260 cm^−1^ in the amide I, amide II, and amide III bands, respectively, which can be attributed to the β-sheet crystallite of silk [30,31]. To examine the effect of sodium carbonate content on the crystallinity of degummed silk, the crystallinity index was computed using Equation (1) and the result is shown in Figure 1C.

The crystallinity index of the degummed silk fibers increased as the sodium carbonate concentration increased to 0.75%, but then remained unchanged. Interestingly, the critical point (0.75% sodium carbonate content) coincides with the onset of fibril formation on SF filaments (Figure 1A(d)). FTIR-ATR utilized in this study reflects the surface of the sample. Consequently, the trend of crystallinity in Figure 2C may be a result of the removal of less crystallized silk. Thus, it appears that SF is composed of silk nanofibrils (which are more crystallized) and inter-fibrillar regions (which are less crystallized). When silk is subjected to over-degumming, the weak inter-fibrillar region is initially damaged. This results in the separation of silk nanofibrils from the SF. Moreover, as the damaged inter-fibrillar region (less crystallized) is eliminated, the silk nanofibrils are peeled off (separated) from the surface of SF to a greater degree. Moreover, as the region of SF with the lowest crystallinity is eliminated, the crystallinity of SF increases. Thus, the less crystallized inter-fibrillar region is removed to the maximum degree at a sodium carbonate concentration of 0.75% resulting in the highest incidence of silk nanofibrils on the surface of SF. Additionally, the increase in crystallinity is stopped at a sodium carbonate concentration of 0.75%.

### 2.2. Preparation and Structural Characteristics of Silk Nanofibril

It is not effective to generate fibrils on SF fiber using only a chemical method (degumming treatment). Therefore, ultrasonication was applied to over-degummed silk to effectively produce more fibrils on the SF filament. Because the greatest number of silk fibrils were generated at 0.75% sodium carbonate, the silk fibers degummed with 0.75% aqueous solution of sodium carbonate were subjected to the ultrasonic treatment. The ultrasonication of the over-degummed silk fibers lasted for various intervals between 3 and 12 h. The effect of ultrasonication time on the morphology of degummed silk fiber is depicted in Figure 2B. Regardless of the ultrasonication time, all treated silk fibers exhibited significantly increased silk nanofibrils on the SF fibers, indicating that ultrasonication is highly effective at producing the fibrils.

The production yield of the silk nanofibrils was determined and the result is shown in Figure 2A. The silk nanofibrils that were completely separated from the SF filament in the solution were collected at intervals of 3 h and production yields were calculated. The production yield of silk nanofibril in the first 3 h was 3.6%. However, as the ultrasonication time was increased, the increasing rate of production yield reduced. Accordingly, the total production yield for 12 h was 8.8%. This indicates that silk nanofibrils can be extracted easily at an earlier stage. However, extraction becomes progressively slower as the ultrasonication time increases. This also suggests that the quantity of silk nanofibril produced by this technique (degumming and ultrasonication treatments) is limited. For reference, Lv et al. demonstrated a ~20% yield for silk nanofibrils using urea [27]. Additionally, Ling et al. reported a yield of ~10% using ultrasonic treatment, which is comparable to the current study [25].

Figure 2C illustrates the morphology of silk nanofibrils as a function of ultrasonication duration. The silk nanofibrils obtained by freeze-drying an aqueous suspension of silk nanofibrils exist as fibrous webs. Additionally, the average diameter of the resulting silk fibril ranges between 130 and 200 nm. Although the diameter variation is relatively high, the ultrasonication time appears to have a minimal impact on the mean diameter of the silk fibrils. According to the findings of previous research, different diameters of silk nanofibrils may be obtained depending on the preparation conditions. Using an ultrasonic technique, Zhao et al. reported 20–60 nm thick silk nanofibers [23]. Additionally, Lv et al. obtained silk nanofibrils of diameter 100–600 nm using a urea solution [27]. Meanwhile, Ling et al. produced silk nanofibrils with a diameter of 20 ± 5 nm using HFIP and ultrasonic treatments [25].

To examine the molecular conformation and crystallinity of the silk nanofibrils, the samples were analyzed using FTIR, and the result is shown in Figure 2E. Regardless of the ultrasonication time, all the silk nanofibril samples exhibited IR absorption peaks at the same positions as the degummed silk fibers. Moreover, the IR peak intensity hardly changed with ultrasonication time.

It is interesting to note that the IR absorption peak of the silk nanofibrils at 1265 cm^−1^ becomes more evident in comparison to the degummed silk fibers (Figure 1B). Moreover, comparing the FTIR index of various silks revealed that their crystallinity is quantitatively distinct. As seen in Figure 3, the silk nanofibrils exhibited a significantly high crystallinity index (68.1%). Conversely, the degummed and over-degummed silk fibers exhibited crystallinity indices of 54.6% and 56.7%, respectively. Moreover, the electrospun RSF web exhibited the lowest crystallinity index (50.6%). This indicates that the silk nanofibrils have an exceptionally high degree of crystallinity in comparison to other silk forms. The lowest crystallinity index of the electrospun RSF web is attributable to the disruption of the β-sheet crystallite of SF during the dissolution [32,33,34]. The significantly higher crystallinity of the silk nanofibrils in comparison to the degummed silk fibers (i.e., SF) confirms that the SF (degummed silk) is composed of a more crystallized fibril region and a less crystallized inter-fibrillar region.

Based on the aforementioned results, a mechanism for the occurrence of silk nanofibrils has been proposed (Figure 4). SF (degummed silk, A) is composed of nanofibrils and inter-fibrillar regions. The inter-fibrillar region of the SF becomes disrupted or damaged by chemical and mechanical attacks during the over-degumming process. The disruption of the inter-fibrillar region leads to the separation of the fibrils from the filaments, resulting in the formation of fibrils in over-degummed SF filaments (B). The damaged inter-fibrillar region in over-degummed SF is easily disrupted by ultrasonication, accelerating the occurrence of fibrils on the SF filaments (C), thereby generating the silk nanofibrils (D).

The mechanical force alone (i.e., ultrasonication) has a limit for producing the silk nanofibrils, resulting in a decrease in the yield of silk nanofibrils as the ultrasonication time increases (Figure 2A). Conversely, Lv et al. demonstrated a ~20% yield for silk nanofibrils using urea, as mentioned previously. Thus, it is assumed that the use of urea disrupted more inter-fibrillar regions of SF, resulting in a greater silk nanofibril yield. Therefore, it appears that the use of chemical treatment in addition to mechanical force is necessary to increase the yield of silk nanofibrils.

Considering that (1) the crystallinity index of silk nanofibrils is greater than that of degummed silks and (2) the crystallinity index of degummed silk increased until the first appearance of silk fibril (0.75% sodium carbonate, Figure 1A,C), the silk nanofibrils are significantly more crystalline than the inter-fibrillar region of SF. Thus, the less crystalline inter-fibrillar region of SF can be disrupted or damaged by degumming and ultrasonication, whereas the highly crystalline silk nanofibril maintains its fibrillary structure. The fact that the crystallinity index of silk nanofibril is greater than that of degummed silks may be due to the composition of degummed silk, which consists of a more crystallized nanofibril and less crystallized inter-fibrillar region. Therefore, as the less crystallized inter-fibrillar region is eliminated, the overall crystallinity of over-degummed silk increases, leading to the formation of fibrils on the degummed silk filament.

### 2.3. Structural Characteristics and Mechanical Properties of Silk Nanofibril/RSF Composite Film

As described in the preceding section, highly crystalline silk nanofibrils could be successfully produced by degumming and ultrasonication. To improve the mechanical properties of RSF films, silk nanofibrils were added to RSF in order to fabricate silk nanofibril/RSF composite films in this study. Additionally, the influence of the amount of silk nanofibril addition on the structure and mechanical properties of the composite film was investigated.

Figure 5 shows the FTIR spectra of silk nanofibril/RSF composite films prepared with varying amounts of silk nanofibrils. The composite films exhibited IR absorption peaks at 1620 and 1510 cm^−1^, which were ascribed to the β-sheet crystallite of silk. Intriguingly, regardless of the amount of silk nanofibril addition, the IR spectra and crystallinity index of the films did not change. The formation of crystallites in these composite films is a result of formic acid casting [30,35], and the IR result for RSF film is consistent with previous findings [19,36]. The constant IR spectra and crystallinity index of composite films, regardless of the amount of silk fibril, may be attributable to the fact that the FTIR ATR measurement is sensitive to the surface of the sample and the majority of silk nanofibrils were embedded in the RSF film, as evidenced by the SEM results (Appendix A). In other words, no silk nanofibril was observed on the surface of the silk nanofibril/RSF composite, indicating that the silk nanofibrils are embedded within the RSF.

Recent research has examined the significance of silk nanofibril structure in determining the mechanical properties of silk [37,38,39,40,41,42,43]. The hierarchical structures of silk nanofibrils, which are composed of β-sheet nanocrystals and a semi-amorphous region, are directly correlated with the remarkable mechanical properties. The ordered H-bond networks in β-sheet nanocrystals and the hidden length in semi-amorphous regions enable silk fibrils to achieve exceptional strength and great extensibility, respectively. When silk fibers undergo deformation, the β-sheet nanocrystal gradually extends, and the concealed length in semi-crystalline regions begins to unravel. Thus, the orientation of macromolecular chains and the arrangement of H-bonds in β-sheet nanocrystals are altered. Although individual H-bonds are weak, when they form networks, they work cooperatively to develop the mechanical properties of silk. During the rupture process, the macromolecular chains are reoriented and loads are transferred between chains by forming interlocking regions. Therefore, H-bond networks in beta-sheet nanocrystals provide cohesion between polypeptide strands, which directly aids in the stretching of amorphous regions. Eventually, the β-sheet nanocrystals are ruptured at large loads and deformation.

Figure 6 illustrates that the greater the number of silk nanofibrils in RSF films, the greater the number of H-bond networks in the system, which contributes directly to the mechanical properties of silk. Accordingly, the tensile strength increased from 37 MPa to 59 MPa (1.6 fold) when the amount of silk nanofibrils was increased to 10 wt.%. Subsequently, it decreased to 38 MPa. This could be due to an abundance of silk fibrils in the system. When the silk nanofibrils are embedded too much, the semi-amorphous region may not be able to fully stretch during deformation, resulting in rupture under smaller loads and deformation. In the case of elongation, the proportion of silk nanofibrils increased steadily. As previously explained, the greater the number of H-bond arrays in the beta-sheet nanocrystal of silk fibril, the greater its stretching capacity. Therefore, it permits the RSF to expand significantly. With the addition of 15% silk nanofibrils, the elongation of RSF film could be significantly increased (3.4 fold). Accordingly, the initial Young’s modulus of the composite films demonstrated a similar trend to the tensile strength observed. These results indicated that silk nanofibrils can effectively improve the mechanical properties of RSF films.

## 3. Materials and Methods

### 3.1. Preparation of Silk Fibrils

To examine the effect of degumming on silk fiber, *Bombyx mori* silkworm cocoons were degummed using boiling sodium carbonate (0.1–1.5% (*w*/*v*)) and sorbitan monostearate (0.1% (*w*/*v*)) solution for 1 h. The liquor ratio was 1:25. Following degumming, the degummed silk fibers were dried at 105 °C. The degummed SF fibers were soaked in a bath of purified water. The liquor ratio was 1:200. Using a water purification system (RO50, Hana Science, Seoul, Korea) with a reverse osmosis membrane, the water was purified. Ultrasonic power was applied to the degummed silk fiber suspension for 3–12 h using an ultrasonicator (Sonomasher, Ulsso Hitech, Cheongwon, Korea) with 560 W of power at 20 kHz frequency. After three hours of ultrasonication, the degummed silk fibers were extracted and transferred to a new bath of purified water. This procedure was carried out four times. The degummed silk fiber was ultrasonically treated for 3, 6, 9, and 12 h. Silk nanofibril was prepared by freeze-drying the remaining silk nanofibril suspensions (Figure 7).

### 3.2. Preparation of the Silk Nanofibril/RSF Composite Film

The preparation method for RSF has been described elsewhere [28,44]. Briefly, to prepare SF, *Bombyx mori* silkworm cocoons were degummed in a boiling aqueous solution containing 0.3% (*w*/*v*) sodium oleate and 0.2% (*w*/*v*) sodium carbonate for 1 h. The liquor ratio was 1:25. After degumming, the cocoons were thoroughly rinsed with purified water and then dried. The degummed silk (i.e., SF) was dissolved at 85 °C for 3 min in a ternary solvent containing CaCl_2_/H_2_O/EtOH (1/8/2 molar ratio). The liquor ratio was 1:20. The RSF aqueous solutions were obtained by dialyzing the dissolved SF solutions in a cellulose tube (molecular weight cut off = 12,000–14,000 Da) against circulating purified water for 5 days at room temperature. The RSF aqueous solutions were dried to obtain the RSF powder was produced by dehydrating RSF aqueous solutions. The RSF was dissolved in 98% (*v*/*v*) formic acid for 3 h to produce a 1% RSF (*w*/*w*) formic acid solution, which was then filtered twice through a non-woven fabric. The various concentrations of the silk nanofibrils (5 wt.%, 10 wt.%, and 15 wt.% by weight of RSF) were submerged in a formic acid solution. An ultrasonicator was then used to disperse silk nanofibril uniformly in formic acid for 30–20 min. Using a mechanical stirrer, the RSF formic acid solution and the silk nanofibrils-formic acid suspension were combined and stirred for 10 min. The 50 mL silk nanofibril/RSF mixed solution was poured into a 9 cm petri dish and dried under a hood at 25 °C to create the silk nanofibril/RSF composite film.

### 3.3. Preparation of the Electrospun RSF Web

The electrospun RSF web was prepared to determine its crystallinity index. The electro-spinning procedure for RSF solutions has been described in detail previously [32,45,46,47]. The RSF powder was dissolved in formic acid (98%) and filtered through a polyester non-woven material to produce 10% regenerated SF dope solutions. These solutions were loaded into plastic syringes with a 21-gauge stainless steel needle (inner diameter = 0.495 mm) at the tip. Subsequently, the electro-spinning process was performed at 14 kV, with a tip-to-collector distance of 20 cm.

### 3.4. Measurement and Characterization

The following equation was used to calculate the degumming ratios: Degumming ratio (%) = (1 − dry mass of degummed cocoons/dry mass of native cocoons) × 100 [48].

The dry mass of each cocoon was measured using a moisture analyzer (XM60, Precisa, Swiss). Subsequently, a Fourier-transform infrared (FTIR, Nicolet 380, Thermo Fisher Scientific, Waltham, MA, USA) spectrometer operating in the attenuated total reflection (ATR) mode was utilized to analyze the molecular conformation and crystallinity index of the silk samples [49,50,51,52,53]. The crystallinity index was calculated from the FTIR spectrum as the intensity ratio of 1265 cm^−1^ and 1235 cm^−1^ using Equation (1)
(1)$$\mathrm{Crystallinity index}\;(\%)=\frac{A_{{1265\mathrm{cm}}^{-1}}}{A_{{1235\mathrm{cm}}^{-1}}}\times 100$$

A_1235cm_^−1^: Absorbance at 1235 cm^−1^

A_126__5cm_^−1^: Absorbance at 1265 cm^−1^

The morphology of the silk samples was analyzed using field emission electron microscopy (FE-SEM, S-4800, Hitachi, Japan). Accordingly, the samples were coated with Pt-Pd prior to FE-SEM imaging [54,55,56]. The production yield was calculated using the following equation:

Production yield (%) = (Weight of the produced silk nanofibril/Weight of the degummed silk fibers) × 100.

To evaluate the mechanical properties of the regenerated silk nanofibril/RSF composite films, tensile strength, tensile elongation, and the initial Young’s modulus were measured using a Universal Test Machine (OTT-03, Oriental TM, Ansan, Korea) [36,57,58,59]. The tensile tests were performed using a 3 kgf load cell at an extension rate of 10 mm/min. The length and width of the prepared film samples were 50 mm and 5 mm, respectively. The gauge length was 30 mm. All samples were preconditioned at 20 °C and 65% relative humidity. Additionally, seven distinct films were used for each silk composite sample to carry out the measurements.

## 4. Conclusions

Highly crystalline silk nanofibrils were successfully fabricated in the current study using degumming and ultrasonication treatments. The over-degumming procedure causes damage to SF, resulting in the separation of silk nanofibrils on the surface of SF filaments. Accordingly, the ultrasonication treatment could be utilized effectively to accelerate the formation of silk fibrils on the SF filaments, thereby yielding the silk nanofibrils. The as-obtained silk nanofibrils exhibited a significantly greater degree of crystallinity than conventional SFs. Moreover, the addition of the silk nanofibrils increased the tensile strength and elongation of the RSF film by 1.6 and 3.4 times, respectively, suggesting that the highly crystalline silk nanofibrils can be effectively utilized as a reinforcing material to enhance the mechanical properties of RSF films.

## Figures and Tables

**Figure 1 ijms-23-11344-f001:**
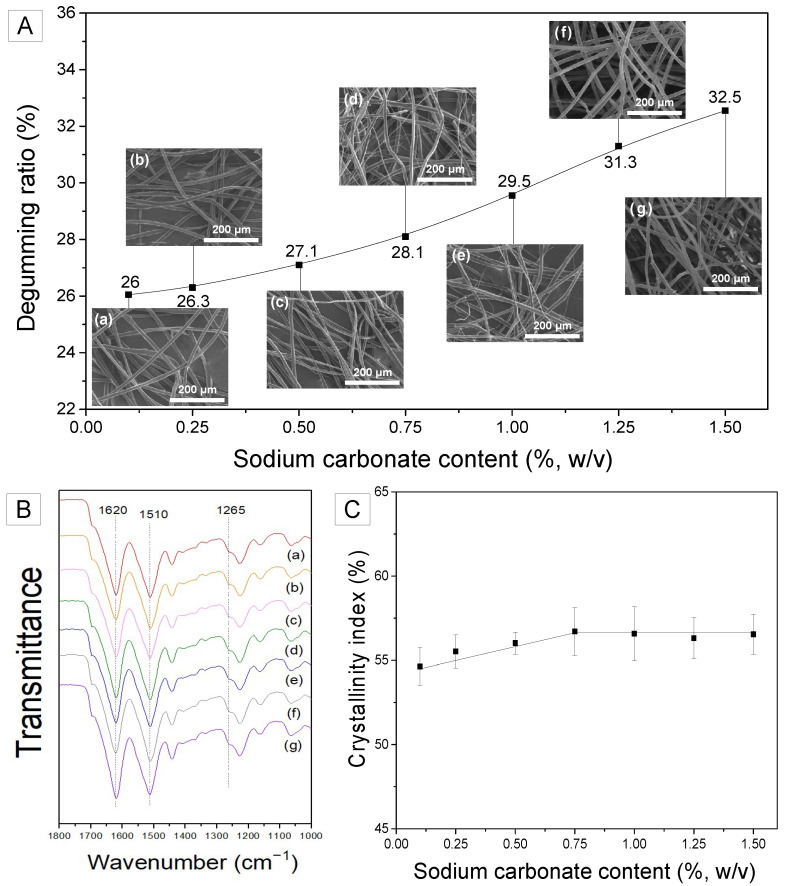
(**A**) Degumming ratio and FE-SEM images, (**B**) FTIR spectra, and (**C**) crystallinity index of silk fibers degummed using various sodium carbonate contents (*w*/*v*); (**a**) 0.1%, (**b**) 0.25%, (**c**) 0.5%, (**d**) 0.75%, (**e**) 1%, (**f**) 1.25% and (**g**) 1.5%.

**Figure 2 ijms-23-11344-f002:**
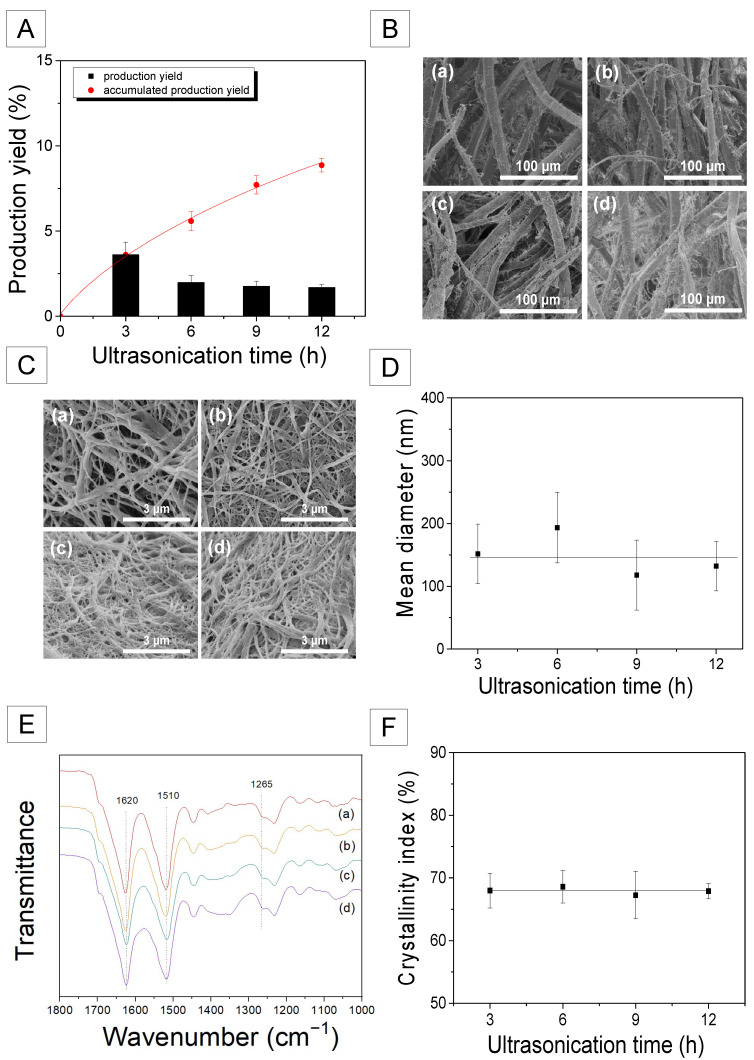
(**A**) Production yields of silk nanofibril, (**B**) FE-SEM images of the degummed and ultrasonicated silk fibers, (**C**) FE-SEM images of the silk nanofibrils, (**D**) Mean diameter of the silk nanofibrils, (**E**) FTIR spectra, and (**F**) crystallinity index of the silk nanofibrils produced at various ultrasonication times; (**a**) 3 h, (**b**) 6 h, (**c**) 9 h, and (**d**) 12 h.

**Figure 3 ijms-23-11344-f003:**
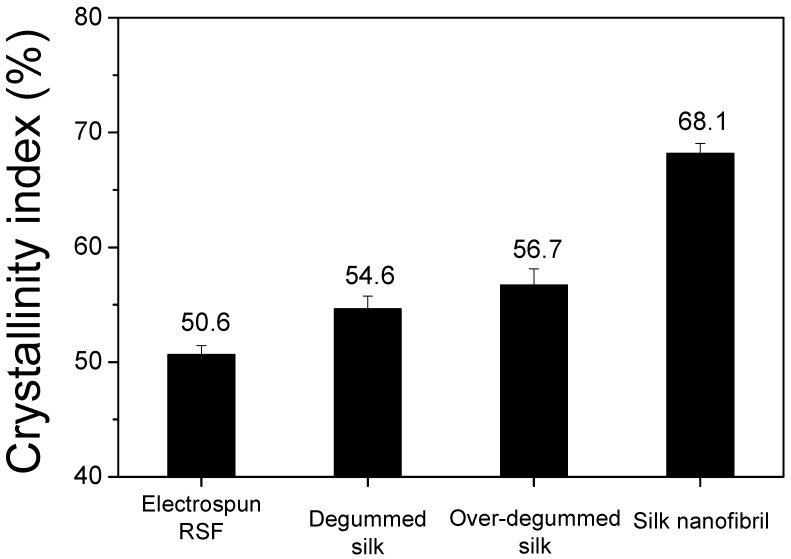
Comparison of the crystallinity indices of various types of silk; electrospun RSF, degummed silk (0.25% sodium carbonate), over-degummed silk (0.75% sodium carbonate) and silk nanofibril.

**Figure 4 ijms-23-11344-f004:**
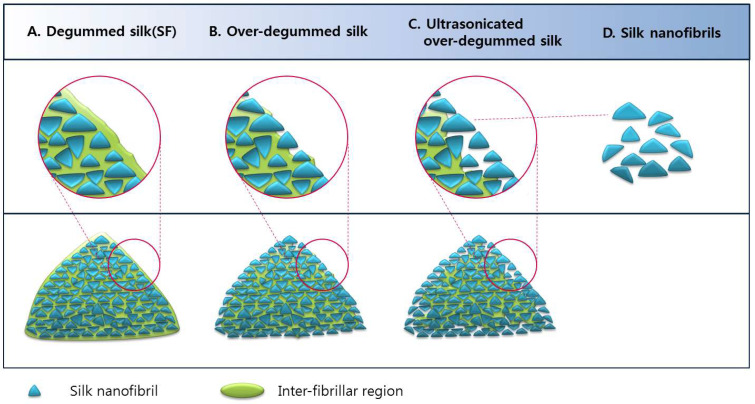
Schematic representation of the structure of SF and occurrence of silk nanofibrils on SF.

**Figure 5 ijms-23-11344-f005:**
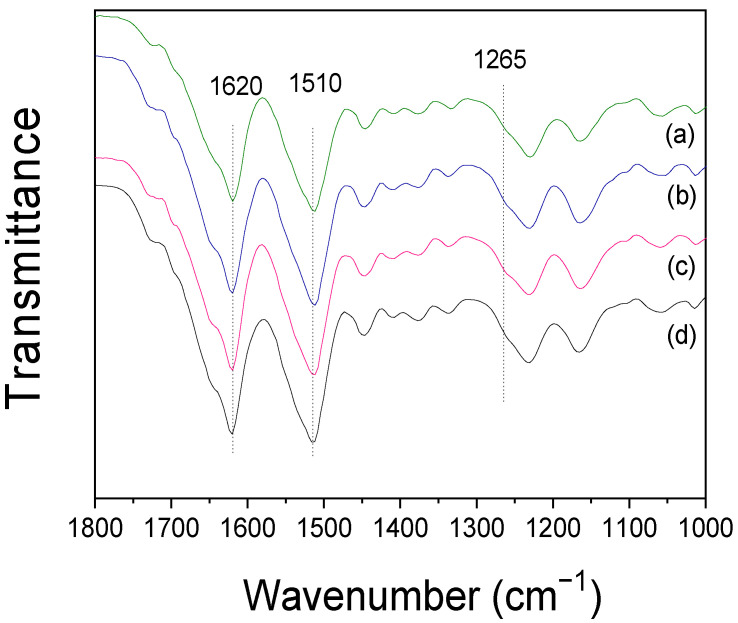
FTIR spectra of the silk nanofibril/RSF composite film containing different amounts of silk nanofibrils; (**a**) 0%, (**b**) 5%, (**c**) 10%, and (**d**) 15%.

**Figure 6 ijms-23-11344-f006:**
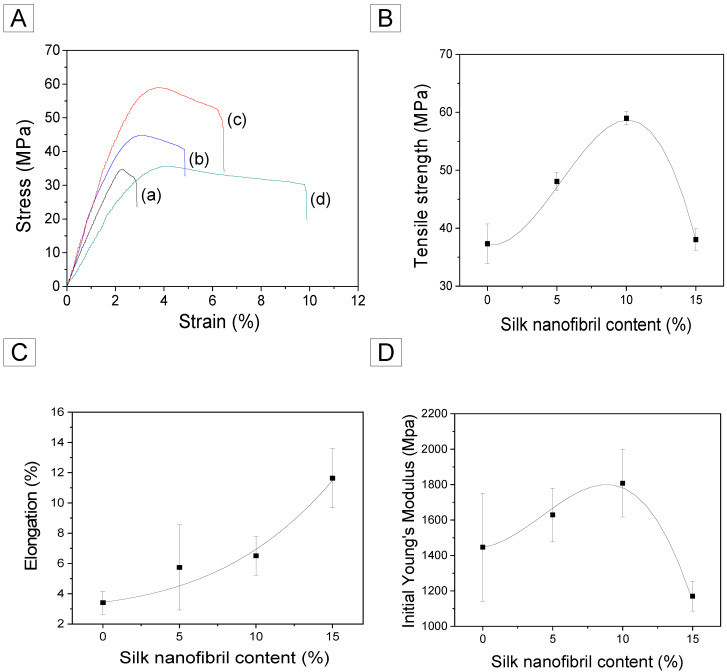
(**A**) Representative stress–strain curve: (**a**) 0%, (**b**) 5%, (**c**) 10%, and (**d**) 15%, (**B**) tensile strength, (**C**) elongation, and (**D**) initial Young’s modulus of the silk nanofibril/RSF composite films with different amounts of silk nanofibrils.

**Figure 7 ijms-23-11344-f007:**
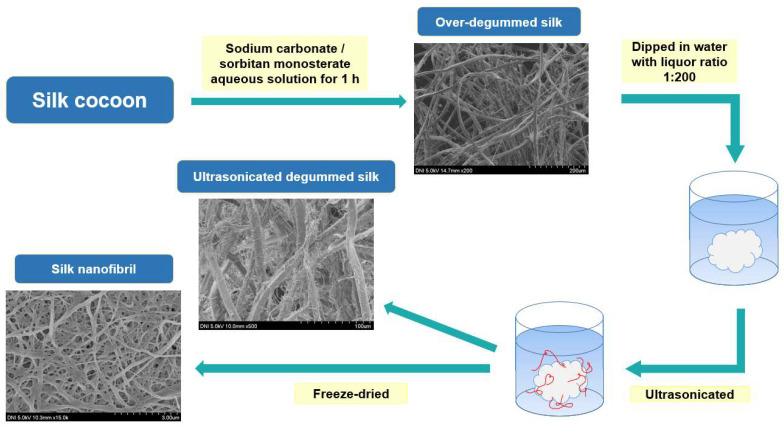
Schematic representation of the fabrication of silk nanofibrils.

## Data Availability

The data presented in this study are available on request from the corresponding author.

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
