# Peer review of "Preparation of Highly Crystalline Silk Nanofibrils and Their Use in the Improvement of the Mechanical Properties of Silk Films"

_ijms, 2022, doi:10.3390/ijms231911344_

Round 1
Reviewer 1 Report
Lee et al. describe a method of preparing silk fibroin nanofibers based on degumming and physical post-treatments (ultrasonication and freeze-drying). The approach is interesting for improving the mechanical properties of silk film, but similar to those previously reported in the literature (as the authors mention in the introduction). I would encourage the authors to consider testing the properties of the films in the potential application.
Although the results are well structured, and the discussion seems coherent, I cannot recommend this paper for publication in IJMS, and I think it would be more suitable for a more specific scopus journal. More specifically:
1. In the introduction, the authors emphasized that foreign polymers may deteriorate the biocompatibility of the silk. This statement is not totally true, many biocompatible polymers (both chemical and recombinant) have been used without compromising biocompatibility:
10.1002/jbm.a.33149, https://doi.org/10.1016/j.bioactmat.2020.09.003
2. I would suggest the authors to substitute the letters a, b, c, and d in figures 2, 5, and 6 with the timepoint or percentages that they represent.
3. Figure 5B should be in the supporting information. It provides limited information on the topography of the sample. I would suggest images at higher magnifications, complemented by AFM analysis.
4. Check all the formulas in the manuscript. Some of them are not in the appropriate format.
5. In the preparation of the electrospun RSF web. The authors should give more details about the preparation of the solutions. Which filter was used? Which pore size?
Author Response
I have read the comments of reviewer carefully and answered point by point.
Please see the attached file of our response. Thank you.

Reviewer 2 Report
I would like to recommend this interesting manuscript for publication on International Journal of Molecular Sciences, while several small problems should be noted:
1. Most of the references in the article are too old. Please introduce some relevant studies published in recent years in the Introduction.
2. Figure 1A, there should be a space between the “Sodium carbonate content” and “(% (w/v))”.
3. Figure 1C, there should be a space between the “Sodium carbonate content” and “(% (w/v))”.
4. Figure 1A and C, the “(% (w/v))” could be revised as “(%, w/v)”.
5. Page 12, what does the “[54-56]” with red color mean?
6. Biocompatibility is an important part of coating evaluation of implanted devices. The authors should evaluate the biocompatibility of the coating according to the application needs. For the operation of biological evaluation in the following references, the author can choose some characterization at his own discretion, if possible.
(1) Y. Yu, S.J. Zhu, H.T. Dong, et al., Journal of Magnesium and Alloys 2021, https://doi.org/10.1016/j.jma.2021.06.015.
(2) S. Guan, K. Zhang, L. Cui, et al., Biomaterials Advances 133 (2022) 112604.
Author Response
I would like to recommend this interesting manuscript for publication on International Journal of Molecular Sciences, while several small problems should be noted:
Comment 1. Most of the references in the article are too old. Please introduce some relevant studies published in recent years in the Introduction.
Answer 1. Thank you for the good comment. We replaced the old articles to new references in the introduction section. Also, we have read the manuscript carefully again and revised minor mistakes in the manuscript.
Comment 2. Figure 1A, there should be a space between the “Sodium carbonate content” and “(% (w/v))”..
Answer 2. Thank you for the good comment. We revised it as commented
Comment 3. Figure 1C, there should be a space between the “Sodium carbonate content” and “(% (w/v))”..
Answer 3. Thank you for the good comment. We revised it as commented
Comment 4. Figure 1A and C, the “(% (w/v))” could be revised as “(%, w/v)”.
Answer 4. Thank you for the good comment. We revised it as commented
Comment 5. Page 12, what does the “[54-56]” with red color mean?
Answer 5. We are sorry. There is no meaning. We just forgot changing the color. Finally, we changed it to black color. Thank you for the comment.
Comment 6. Biocompatibility is an important part of coating evaluation of implanted devices. The authors should evaluate the biocompatibility of the coating according to the application needs. For the operation of biological evaluation in the following references, the author can choose some characterization at his own discretion, if possible.
(1) Y. Yu, S.J. Zhu, H.T. Dong, et al., Journal of Magnesium and Alloys 2021, https://doi.org/10.1016/j.jma.2021.06.015.
(2) S. Guan, K. Zhang, L. Cui, et al., Biomaterials Advances 133 (2022) 112604.
Answer 6. Thank you for the good comment. But, the evaluation of biocompatibility is out of scope of this study. In the present study, we concentrated the fabrication and structural examination of silk nanofibrils and it use in the silk nanofibrils/regenerated silk fibroin film. Please understand this. According to your comment, we will refer the two papers you suggested, when we conduct the evaluation of biocompatibility in the subsequent studies.
Reviewer 3 Report
The authors present highly crystalline silk nanofibrils, which are successfully extracted from silk by degumming and ultrasonication. With a proper addition of silk nanofibrils into RSF films, the breaking strength and elongation of the RSF films increase by 1.6 fold and 3.4 fold, respectively. The mechanical properties of the resulting composite RSF films are outstanding, suggesting that highly crystalline silk nanofibrils are an excellent reinforcing material for RSF films. The manuscript is recommended for publication in International Journal of Molecular Sciences. Some minor revisions are suggested below.
1. The manuscript describes that “as the amount of degumming agent was increased, the surface of the SF filament became rougher, indicating that the fibrils separated from the SF filaments”. But in Figure 1(A), it’s hard to distinguish the difference between the samples treated with different sodium carbonate content. SEM images with higher magnification are suggested.
2. The authors claim that the crystallinity index of degummed silk fibers increased as the sodium carbonate concentration increased to 0.75%, but then remained unchanged, resulting from the removal of less crystallized silk. However, Figure 3 shows that the crystallinity index of silk nanofibrils is larger than over-degummed silk fibroin fibers. It seems that there is a conflict.
3. Some things are missing in Equation 1.
4. In Figure 5A, the IR spectra and crystallinity index of the films don’t change regardless of the amount of added silk nanofibrils. The authors explain that this phenomenon may be attributed to the fact that FTIR ATR measurements are sensitive to the surface of the samples while the majority of silk nanofibrils are embedded in the RSF films. Could the transmission mode of FTIR solve this problem?
5. Two recent papers on protein fibers are suggested: CCS Chemistry 2020, 2, 1669–1677, and Engineering, 2022, 14, 100-112.
Author Response
The authors present highly crystalline silk nanofibrils, which are successfully extracted from silk by degumming and ultrasonication. With a proper addition of silk nanofibrils into RSF films, the breaking strength and elongation of the RSF films increase by 1.6 fold and 3.4 fold, respectively. The mechanical properties of the resulting composite RSF films are outstanding, suggesting that highly crystalline silk nanofibrils are an excellent reinforcing material for RSF films. The manuscript is recommended for publication in International Journal of Molecular Sciences. Some minor revisions are suggested below.
Comment 1. The manuscript describes that “as the amount of degumming agent was increased, the surface of the SF filament became rougher, indicating that the fibrils separated from the SF filaments”. But in Figure 1(A), it’s hard to distinguish the difference between the samples treated with different sodium carbonate content. SEM images with higher magnification are suggested.
Answer 1. Thank you for the comment. However, when we magnify the manuscript, we can see the fibrils separated from the SF filaments easily. When we increase the magnification of SEM further, we cannot see the morphology of overall degummed silk fibers. That is why we use this magnification. Please understand this.
Comment 2. The authors claim that the crystallinity index of degummed silk fibers increased as the sodium carbonate concentration increased to 0.75%, but then remained unchanged, resulting from the removal of less crystallized silk. However, Figure 3 shows that the crystallinity index of silk nanofibrils is larger than over-degummed silk fibroin fibers. It seems that there is a conflict.
Answer 2. Thank you for the comment. I think the reviewer confused with less crystallized silk and silk nanofibril.
We mentioned in the manuscript that SF is composed of silk nanofibrils (which are more crystallized) and inter-fibrillar regions (which are less crystallized). When silk is subjected to over-degumming, the weak inter-fibrillar region is initially damaged. This results in the separation of silk nanofibrils from the SF. Moreover, as the damaged inter-fibrillar region (less crystallized) is eliminated, the silk nanofibrils are peeled off (separated) from the surface of SF to a greater degree. Moreover, as the region of SF with the lowest crystallinity is eliminated, the crystallinity of SF increases. Thus, the less crystallized inter-fibrillar region is removed to the maximum degree at a sodium carbonate concentration of 0.75% resulting in the highest incidence of silk nanofibrils on the surface of SF. Additionally, the increase in crystallinity is stopped at a sodium carbonate concentration of 0.75%.
Also, we wrote in the manuscript that Considering that 1) the crystallinity index of silk nanofibrils is greater than that of degummed silks and 2) the crystallinity index of degummed silk increased until the first appearance of silk fibril [0.75% sodium carbonate, Figure 1A, 2C1C)], the silk nanofibrils are significantly more crystalline than the inter-fibrillar region of SF. Thus, the less crys-talline inter-fibrillar region of SF can be disrupted or damaged by degumming and ul-trasonication, whereas the highly crystalline silk nanofibril maintains its fibrillary struc-ture. The fact that the crystallinity index of silk nanofibril is greater than that of degummed silks may be due to the composition of degummed silk, which consists of a more crystallized nanofibril and less crystallized inter-fibrillar region. Therefore, as the less crystallized inter-fibrillar region is eliminated, the overall crystallinity of over-degummed silk increases, leading to the formation of fibrils on the degummed silk fila-ment.
That is, the silk nanofibril is highly crystallized material as shown in Figure 3. Also, the crystallinity of degummed silk is increased at 0.75% sodium carbonate content, because the less crystallized inter-fibrilar region is removed until 0.75% sodium carbonate content
Also, we added “silk nanofibril” and “inter-fibrilar region” in the Figure 4 for a better understanding of a reader.
Comment 3. Some things are missing in Equation 1.
Answer 3. Thank you for the comment. When the MDPI edit team made the present format of manuscript, I think the small image was moved out during the editting. We modified it not to move anymore. Thank you.
Comment 4. In Figure 5A, the IR spectra and crystallinity index of the films don’t change regardless of the amount of added silk nanofibrils. The authors explain that this phenomenon may be attributed to the fact that FTIR ATR measurements are sensitive to the surface of the samples while the majority of silk nanofibrils are embedded in the RSF films. Could the transmission mode of FTIR solve this problem?
Answer 4. Thank you for the comment. That is a good idea. Unfortunately, we have done these experiments several years ago. Therefore, we cannot conduct what the reviewer suggested. But, that is a good idea. Thus, we will use FTIR with transmittance mode, when we study silk nanofibrils composites in the future.
Comment 5. Two recent papers on protein fibers are suggested: CCS Chemistry 2020, 2, 1669–1677, and Engineering, 2022, 14, 100-112.
Answer 5. Thank you for suggesting good papers. We have read the two papers and added “Engineering, 2022, 14, 100-112” as a new reference in this manuscript. We think “CCS Chemistry 2020, 2, 1669–1677” is less relevant with this study.
Round 2
Reviewer 1 Report
I thank the authors for reviewing the manuscript. After the changes were made, I believe that the quality of the manuscript has been improved.